# Molecular Mechanisms of Persistence in Protozoan Parasites

**DOI:** 10.3390/microorganisms11092248

**Published:** 2023-09-07

**Authors:** Asfiha Tarannum, Cristian Camilo Rodríguez-Almonacid, Jorge Salazar-Bravo, Zemfira N. Karamysheva

**Affiliations:** Department of Biological Sciences, Texas Tech University, Lubbock, TX 79409, USA; atarannu@ttu.edu (A.T.); rod65531@ttu.edu (C.C.R.-A.); j.salazar-bravo@ttu.edu (J.S.-B.)

**Keywords:** protozoa persisters, metabolome, translatome, Apicomplexa

## Abstract

Protozoan parasites are known for their remarkable capacity to persist within the bodies of vertebrate hosts, which frequently results in prolonged infections and the recurrence of diseases. Understanding the molecular mechanisms that underlie the event of persistence is of paramount significance to develop innovative therapeutic approaches, given that these pathways still need to be thoroughly elucidated. The present article provides a comprehensive overview of the latest developments in the investigation of protozoan persistence in vertebrate hosts. The focus is primarily on the function of persisters, their formation within the host, and the specific molecular interactions between host and parasite while they persist. Additionally, we examine the metabolomic, transcriptional, and translational changes that protozoan parasites undergo during persistence within vertebrate hosts, focusing on major parasites such as *Plasmodium* spp., *Trypanosoma* spp., *Leishmania* spp., and *Toxoplasma* spp. Key findings of our study suggest that protozoan parasites deploy several molecular and physiological strategies to evade the host immune surveillance and sustain their persistence. Furthermore, some parasites undergo stage differentiation, enabling them to acclimate to varying host environments and immune challenges. More often, stressors such as drug exposure were demonstrated to impact the formation of protozoan persisters significantly. Understanding the molecular mechanisms regulating the persistence of protozoan parasites in vertebrate hosts can reinvigorate our current insights into host–parasite interactions and facilitate the development of more efficacious disease therapeutics.

## 1. Introduction

The emergence of persister-like cells (populations exhibiting persistence-like traits) as a novel arsenal in the face of drug and host immune response has brought about a paradigm shift in our understanding of disease pathology and drug resistance. Events of therapeutic failure and drug resistance are common case scenarios in the context of cancer and infectious diseases. Resistance to antibiotics by producing beta-lactamases and altering drug targets confer the ability to withstand the chemotoxic environment to a selective group of mutant bacterial populations [1]. Chemotherapeutic drug resistance by cancer cells, resistance to chloroquine, antifolates, arteminisins by *Plasmodium* spp., resistance to the tuberculosis drug isoniazid (INH: isonicotinic acid hydrazide) through decreased drug activation by *Mycobaterium* spp., tetracycline resistance through bacterial drug efflux transporters, resistance to pentavalent antimonials, amphotericin B, and miltefosine by *Leishmania* spp., benznidazole resistance by *Trypanosoma cruzi*, pyrimethamine (PYR) and sulfadiazine (SDZ) resistance by *Toxoplasma gondii*, and many more examples of drug resistance acquired by pathogens are testament to the burgeoning concept of persistence in the wake of drug resistance, treatment failure and relapse of chronic diseases [2].

Another distinct mechanism that contributes to the survival of parasites in the host is achieved through the presence of persisters. Persisters were first interpreted as dormant, non-dividing cells contributing to the survival of a small number of *Staphylococcus* spp. in a penicillin treatment study by Joseph Bigger in 1944 [3]. Moreover, persister cells differ significantly from resistant cells in that the former can obviate stress in a non-replicative state, whereas the latter can proliferate under stress. Once stress is removed from the environment, the persisters return to their previous state and resume their active forms of life [4] (Figure 1). This dormancy/latency of the persisters might be triggered spontaneously under stress conditions, and the pathogens survive treatments without selection of genetically heritable mutations [4,5]. The dormancy state is accompanied by a slow metabolism and reduced mRNA translation, the most common features shared by all protozoa persisters (Figure 1B). Thus, the ability of pathogenic microorganisms to enter a quiescent/dormant state contributes to treatment failures and prolonged infections.

Today, multidrug resistance has emerged among Gram-positive and Gram-negative bacteria, rendering conventional antimicrobials, to some extent, ineffective; this capacity of pathogenic bacteria to withstand therapeutic intervention, otherwise known as antimicrobial drug resistance, is swiftly emerging as a severe clinical concern [6]. On the other hand, persistence is a survival mechanism that bacteria use as a defense against various harmful environmental factors. It consists in the appearance of a latent state with much reduced metabolic activity that allows the microorganisms to withstand the exposure to antimicrobials [3]. There is evidence that numerous bacterial species, including human pathogens like *Pseudomonas aeruginosa* and *Mycobacterium tuberculosis*, as well as *Lactococcus lactis* and *Escherichia coli*, exhibit subpopulations of persistent cells [7,8].

In addition to disease reactivation, these persistent cells stimulate the concomitant immunity, a protective response against recurrent infections [9]. Dormancy in *E. coli* persisters is regulated through reduced protein synthesis upon treatment with β-lactam antibiotics [10,11]. Among the multiple mechanisms involved in bacterial persistence, the best studied is that involving the Toxin–Antitoxin (TA) system [12]. The toxin, which is liberated by the degradation of the antitoxin, induces the persister-like state by suppressing DNA replication, transcription, and translation. When bacteria are exposed to an antibiotic or stress, the SOS reaction is activated, while cell respiration and energy production are inactivated, thereby inhibiting protein synthesis [13,14]. 

Likewise, chemotherapy-tolerant persister-like cells appear in cancer progression. Dormant tumor subpopulations continue to be a significant obstacle to the success of cancer treatments. Recent studies showed that specific cell subsets within isogenic drug-sensitive cancer populations can adopt a “persister” condition that enables them to endure long-term therapies. The development of a variety of mechanisms that support the persistence of cancer cells was observed in the last decades, including epigenetic, transcriptional, and translational activities that frequently coexist and are not mutually exclusive. Persistent cells use four primary, non-exclusive tactics to avoid drug treatments, which are (1) reducing cell growth, (2) modifying cell metabolism, (3) changing cell identity, and (4) hijacking the microenvironment [10]. The initial drug treatment leads to tumor regression, which is followed by a stable timespan of “minimal residual disease” (MRD) in which a small population of cancer cells exists within tumors that are no longer regressing. This population of drug-tolerant persisters gives rise to a fully resistant clone [11]. Cancer persistence, however, is linked to non-genetic changes that are associated with a non- or slow-proliferation status, a reaction that was first noticed in bacteria after antibiotic treatment, which means that cancer persistence is not caused by cancer cell genetic variants [15]. 

Protozoa pathogens cause life-threatening conditions through drug resistance and persistence in their vertebrate hosts. In tropical regions, diseases including malaria, trypanosomiasis, and leishmaniasis are widely prevalent. The survival of protozoa parasites can be achieved by two major molecular mechanisms: through the acquisition of drug resistance or by entering a dormant state (Figure 1). These two mechanisms are very distinct: when dormant cells return to the proliferative state upon the drug withdrawal, they will be sensitive to the drug. In contrast, drug-resistant cells are highly proliferative in the presence of the drug, and their resistance could be driven in part by acquired genetic mutations. The continuous failure of antimalarial and other drugs, the sophisticated immune evasion strategies employed by protozoa parasites, and the widespread resistance to antimonial drugs by *Leishmania* parasites in the tropics, among other mechanisms through which these microorganisms avoid drugs’ effects, leaves us with no choice but to explore new methodologies to comprehend pathogen biochemistry and predict drug resistance mechanisms and immune evasion. 

Protozoan parasites are eukaryotic pathogens that cause crucial health and socio-economic concerns. They are responsible for the neglected tropical diseases (NTD) prevalent in tropical regions, among which malaria, trypanosomiasis, and leishmaniasis are of great significance. An annual estimate of 1.1 million deaths was reported as an aftermath of the fatalities brought upon by these parasitic diseases [16]. Presently, almost 1 billion people reside in areas endemic to leishmaniasis. More than 1 million new cases of cutaneous leishmaniasis (CL) and 30,000 new cases of visceral leishmaniasis (VL) known as kala-azar (the most severe form of the disease) are reported annually [17]. In 2021, 619,000 deaths due to malaria caused by *Plasmodium* spp. were estimated by the WHO [18]. Also, nearly 9100 people lost their lives in 2010 due to African trypanosomiasis disease, caused by *Trypanosoma brucei*, which affected 20,000 individuals [19]. There is mounting evidence that *Trypanosoma* and *Leishmania* spp. may have persister forms contributing towards relapse of the diseases they cause [20,21]. Persistence appears to be quite common in protozoan parasites, occurring in a spontaneous manner or in reactions to environmental cues and often leading to immune evasion and parasite survival to any treatment procedure [22]. The hypnozoite stage of *Plasmodium* spp., the bradyzoite stage of *Toxoplasma* spp., the non-dividing amastigote stages of *Trypanosoma* spp. and *Leishmania* spp. are of particular interest, since in these states, the parasites survive to treatments through persistence mechanisms. 

While cancer and bacterial persistence have so far gained significant attention, protozoan persistence has not been thoroughly explored. The increasingly alarming recent cases of failure in the treatment of protozoa infections have compelled us to explore the biology of persister cells [23]. In this review, we delve deep into the role of these unusual cells in the recalcitrance of protozoan diseases, the molecular/environmental events that might trigger persistence, and the molecular mechanisms of protozoan persistence through the analysis of host –pathogen interactions and immune evasion strategies adopted by the persisters. We also explore persistence from the metabolome, transcriptome, and translatome profiles to examine the importance of different factors in host’s immune system evasion. 

## 2. Mechanisms of Persistence in *Plasmodium* spp.

Malaria, caused by the protozoan parasites of the genus *Plasmodium*, is one of the most dreadful and crucial neglected tropical diseases (NTD) of humans, having clinical and socio-economic impacts on poor tropical and subtropical regions of the world. As of now, more than 200 species of *Plasmodium* have been described, which are known to infect a wide range of vertebrates [18]. Several species of *Plasmodium* can infect humans, including *P. falciparum, P. vivax*, *P. malariae*, *P. ovale*, and *P. knowlesi* [18,24]. The *Plasmodium* spp. are distributed in different regions of the world, including sub-Saharan Africa, Southeast Asia, Latin America, and parts of the Middle East [24]. *P. falciparum* is the most common cause of malaria in sub-Saharan Africa, while *P. vivax* is more commonly found in Asia and Latin America [25]. Globally, there were reportedly 247 million cases of malaria and 619,000 estimated deaths in 2021. Although there are still over 100 countries where malaria is present, sub-Saharan Africa is the most affected area, accounting for over 90% of all deaths, most of which are of children under the age of five years [18]. Vector control and preventive chemotherapy using a combination of antimalarial drugs like Atovaquone/Proguanil (Malarone), Chloroquine, Doxycycline, Mefloquine, Primaquine, and Tafenoquine have been used worldwide to counter malaria [24]. Artemisinin-based combination therapy (ACT) is the best currently available treatment, particularly for *Plasmodium falciparum* [18,24]. *Plasmodium* species have exceptional genetic plasticity enabling them to quickly navigate between their hosts, develop resistance to antimalarial medications, and adapt to environmental changes [26].

The transfer of *Plasmodium* species from one vertebrate host to another relies on an insect vector, primarily the mosquito, where the parasite undergoes sexual reproduction, which is crucial for transmission to the next vertebrate host. Anopheline mosquitoes are the only vectors employed by all five *Plasmodium* species that infect humans. *Plasmodium* sporozoites are introduced into the bloodstream when an infected mosquito bites a person. The sporozoites move towards the liver, their multiplication site, and there transform into merozoites, which are released into the bloodstream, infecting the erythrocytes. Within the host’s red blood cells, the merozoites multiply, causing the cells to burst and release additional merozoites into the bloodstream [24]. The symptoms of malaria manifest chiefly due to the presence of parasites in the blood and might include fever, chills, headache, muscle pains, lethargy, and, in extreme cases, anemia, organ failure, and even death. Due to acquired immunity, many people in malaria-endemic areas carry *Plasmodium* asymptomatically [27]. *Plasmodium* infection during pregnancy can raise the risk of miscarriage, stillbirth, and low birth weight in the baby. It can also be passed down from mother to infant through breastfeeding or childbirth [24]. 

To fully comprehend the phenomenon of persistence of the pathogen, first, we must understand its resistance strategies to combat drugs. *P. falciparum* was found to have the most potent resistance profile among all *Plasmodium* species. Missense mutations in the active site of dihydrofolate reductase (the target enzyme of the antimalarial drugs pyrimethamine and proguanil) have been demonstrated to render *P. falciparum* particularly resistant to antifolate drugs [28,29]. Also, point mutations in the genes *pfcrt* (*P. falciparum* chloroquine resistance transporter) and *pfmdr1* (*P. falciparum* multidrug resistance 1) encoding the transporter proteins CRT and MDR1, respectively, aid in removing the antimalarials from *P. falciparum* [30]. As an aftermath, parasites with these altered genes develop resistance to antimalarials that prevent heme detoxification in the digestive vacuoles. Moreover, an increased *pfmdr1* gene copy number can be linked to resistance to Mefloquine [31]. Mutations in the *P. falciparum* Kelch13 protein (K13), a protein involved in numerous intracellular activities including the endocytosis of hemoglobin, may mediate artemisinin resistance [32]. 

Regarding the immune evasion and persistence of *Plasmodium* parasites, *Plasmodium falciparum* avoids detection by the human host’s immune system by modifying the expression of the *var* gene family and of other variant surface antigens (VSA) during its replication inside red blood cells [33,34]. It has also been reported that a small number of parasites that escape detection in the patient’s bloodstream after an initial treatment can cause disease recrudescence, whereas relapse occurs due to dormant hypnozoites in the liver of patients infected by *P. vivax* and *P. ovale* [35]. Likewise, *Plasmodium* parasite persistence has been documented after the short-term (6 h to 144 h) administration of antimalarial drugs such as artemisinin derivatives, atovaquone, proguanil, pyrimethamine, and mefloquine, as well as after exposure to stressors like cold shock or nutritional deprivation [36,37,38,39,40,41]. Dormancy or persistence is often resistant to perturbations and correlates with the presence of dense chromatin and diminished cytoplasm, corroborating that persistence is the cellular response to adapt to environmental stress [22]. *P. falciparum* cells treated with dihydroartemisinin (DHA) and other artemisinin derivates demonstrated arrested development, followed by the reactivation of persistent populations (0.04–1.3% of parasites) 9 to 20 days after drug withdrawal [40,42]. Artemisinin-treated persistent *Plasmodium* parasites showed lower metabolism, except for fatty acid synthesis and pyruvate pathways, but active apicoplasts (plastid of the phylum Apicomplexa) and mitochondria [43,44]. However, the mitochondria of persister parasites showed changes in morphology and metabolism, as well as reduced mito-nuclear distances when exposed to drugs, which indicated that the mitochondria are essential for the persistence and recovery of dormant parasites [45,46]. An artemisinin-resistant K13 M476I mutant subjected to short-term drug pressure was observed to form quiescent, non-pyknotic rings (a state where chromatin is irreversibly condensed in the nucleus of a cell undergoing necrosis or apoptosis), resulting in parasite recovery upon removing the artemisinin pressure [47]. 

Historically, *P. falciparum* has been recognized as a pathogen contributing to higher morbidity and mortality. However, *P. vivax* is currently accepted as a major obstacle to malaria elimination due to its ability to form long-lasting “sleeper” cells (hypnozoites) in the host liver that emerge weeks, months, or sometimes years after the primary infection [48]. When *P. vivax* sporozoites enter the hepatocyte, they transform into replicating schizonts and induce disease or delay replication and persist as hypnozoites. These hypnozoites exhibit a diverse range of transcriptomic states, spanning a spectrum of phenotypes, from persisting to active hypnozoites. Notably, the persister hypnozoites displayed a differential expression of genes encoding specific cellular RNA-binding proteins (RBP), which supports the notion of post-transcriptional mechanisms governing gene effects. Additionally, an enrichment of genes linked to translational repression was observed, as well as an increase in the expression of proteases, which could favor the digestion of host cytosolic proteins [49]. Likewise, *P. vivax* was shown to alter signaling pathways associated with antioxidant stress, energy metabolism, and the immune response in infected hepatocytes, for example, the decreased expression of genes encoding chemokines in parasitized hepatocytes to evade detection by immune cells [49].

The sexual precursor stage of *Plasmodium* is the intraerythrocytic gametocyte, which once mature, remains dormant until it is taken up by the vector [50,51]. Different studies revealed that exposure to antimalarial drugs can cause an increase in gametocytemia [52,53,54,55]. Primaquine is the only recommended drug by the WHO with gametocytocidal activity [56]. Metabolic features of gametocytes include the mechanism of glucose utilization, i.e., the production of acetate as the principal end-product of glycolysis, as well as the increase in the consumption of lipid moieties, which is accompanied by an upregulation of fatty acid pathways [57]. Translational repression during the life cycle of *Plasmodium* spp. has been investigated in several studies. In female gametocytes, DOZI (development of zygote inhibited), an RNA helicase of the DDX6 class, binds to the ribonucleoprotein complex and represses translation, leading to the storage of untranslated mRNAs that can be translated after fertilization [58,59]. Puf1 and Puf2, RNA-binding proteins that bind to the mRNA 3′ untranslated region, are significantly upregulated in gametocytes [60,61] and sporozoites (Puf2) [59,62]. The phosphorylation of eIF2α in salivary gland sporozoites, mature schizonts, and gametocytes induces translational repression and the formation of stress granules where untranslated mRNAs are stored [63,64]. While translational repression is an important mechanism to control life stages in parasites, it is also a crucial mechanism that allows them to achieve latency and enter the persister’s state. Artemisinin treatment leads to the phosphorylation of eIF2α, inducing translational repression in *Plasmodium* and generating dormant forms [65]. The inhibition of the eIF2α blocks the parasites from entering dormancy and abolishes disease recrudescence after artemisinin treatment. Table 1 summarizes the transcriptomic, metabolomic, and translatomic changes observed in *Plasmodium* persisters. 

## 3. Mechanisms of Persistence in *Toxoplasma* spp. 

*Toxoplasma gondii* is a protozoan parasite that requires the host cells for its survival and can infect warm-blooded vertebrates globally; despite its inability to survive outside the host cell, it is highly transmissible. Around a third of the world’s population is believed to be infected with *Toxoplasma* [69,70]. *Toxoplasma* is a member of the phylum Apicomplexa, which comprises other significant pathogens like *Plasmodium*, *Eimeria*, *Babesia*, *Neospora*, *Sarcocystis*, and *Cryptosporidium* [71,72,73]. After infecting an immunocompetent host, *Toxoplasma* typically causes an asymptomatic acute infection, followed by the formation of intracellular tissue cysts and the establishment of chronic infection. The *Toxoplasma*’s parasite ability to cause disease and spread is linked to its capacity to transform from the rapidly dividing tachyzoite phase to inactive tissue cysts known as bradyzoites [74] These cysts can remain dormant within the host tissues throughout the host’s life, kept in check by the immune system [75]. Infection by *Toxoplasma* is believed to trigger lifelong protective immunity due to the parasite’s cysts persisting over time and rupturing regularly, which induces immune protection against future infections. However, in cases of immune suppression, the cysts can reactivate, and the bradyzoites inside them can transform into proliferating tachyzoites, leading to severe tissue destruction [75]. Immunocompromised individuals, particularly those with HIV or those who underwent allogeneic hematopoietic stem cell transplantation (HSCT), face a significant risk of toxoplasmosis reactivation [76,77,78,79].

According to the Lainson’s theory, for long-term immunity to occur, the *Toxoplasma* parasite must persist in the host’s lifetime by forming bradyzoites housed in intracellular cysts [74]. Because the parasite can differentiate into bradyzoites and form impenetrable cysts, eradicating *Toxoplasma* from the host is currently unattainable [80]. Some drugs, like pyrimethamine plus sulfadiazine, can manage acute toxoplasmosis, but no short-term treatment can eliminate the cysts, which also seem to resist the immune response [73]. *Toxoplasma* can become a chronic infection due to the existence of latent bradyzoite cysts. The fact that bradyzoites can revert to rapidly growing tachyzoites explains why immunocompromised individuals often experience high rates of acute toxoplasmosis [73,80]. Additionally, the bradyzoites found within purified mature cysts from mouse brains displayed intravacuolar mobility [81]. Another study showed that fully developed bradyzoites obtained from mice brains infected with *Toxoplasma* oocysts remained in a growth-arrested phase, i.e., in the G0 stage of cell division, and had uniform 1N DNA content [82]. Thus, only fully developed bradyzoites enter the dormant state. Therefore, acquiring a deeper understanding of the molecular processes that drive bradyzoite development is necessary to pinpoint the persistence mechanisms employed by *Toxoplasma* bradyzoites. 

When the parasite switches to a latent lifestyle by entering the bradyzoite stage, there are changes in its metabolism. Several metabolic enzymes have tachyzoite- and bradyzoite-specific isoforms, such as ENO2/ENO1 and LDH1/LDH2, indicating a precise regulation of metabolism in these two life cycle stages [83]. The significant presence of amylopectin granules containing polysaccharides in the bradyzoites indicates a momentous change in carbohydrate metabolism [83]. Biochemical analyses support this notion, as they determined that bradyzoites lack functional TCA cycle and respiratory chain. This indicates that anaerobic glycolysis likely plays a predominant role during this stage [83]. In comparison, tachyzoites probably utilize mitochondrial oxidative phosphorylation and glycolysis for ATP production. Pyruvate kinase and lactate dehydrogenase activities are significantly elevated in bradyzoites, indicating that lactate production is vital during persistence [83]. An isoform of lactate dehydrogenase (LDH2) specific to bradyzoites was discovered and is considered a specialized enzyme necessary for this persistent form of life [84]. Major features observed in *Toxoplasma* persisters are shown in Figure 2.

Two additional isoforms of crucial glycolytic enzymes specific to bradyzoites were recognized: glucose-6-phosphate isomerase (G6-PI) and enolase-1 (ENO1). The various enolase isoforms have different enzymatic properties and differ in their stability [85]. Enolases may also have a role in transcriptional regulation and the adaptation of glycolysis. Both ENO1 and ENO2 were detected in the cytoplasm and nucleus of the parasite in both bradyzoite and tachyzoite forms [85]. Stage-specific enzymes were also found to be tailored to adjust glycolysis and enable either proliferation or dormancy. For instance, some enzymes involved in the metabolism of oxygen radicals seem more active in bradyzoites, indicating that the cyst form can deal with extended exposure to reactive metabolites [86]. This idea is reinforced by studies reporting upregulated mRNAs encoding different DNA repair enzymes in bradyzoites [86,87]. 

There is ample evidence that the switch to the latent stage is a response triggered by stress, which causes the parasite’s cell cycle to slow down. One of the most frequently used in vivo techniques for inducing bradyzoite differentiation is exposing the parasite to alkaline conditions at a pH of 8.0–8.2 [88]. Numerous other stress-inducing agents have been discovered, such as sodium nitroprusside, which serves as a source of exogenous nitric oxide (NO) and inhibits proteins that are part of the parasite’s mitochondrial respiratory chain [89]. Similarly, medications that disrupt the parasite’s mitochondria stimulate the differentiation of tachyzoites into bradyzoites. Exposure to heat shock and sodium arsenite also causes the upregulation of bradyzoite antigens [79]. A lack of nutrients is a potent stimulus for the development of bradyzoites and can be accomplished through various methods, such as arginine’s deprivation, incubation in an axenic environment, or depletion of pyrimidines in UPRT-deficient parasites after exposure to ambient CO2 levels of 0.03% [90,91,92]. Thus, many types of cellular stress can induce bradyzoite formation. Bradyzoites that have been produced in vitro will rapidly convert back to proliferating tachyzoites when the stressor used to differentiate the parasites is removed. 

These research findings emphasize the concept that cellular stress is a crucial factor in both initiating and preserving the encysted form of the parasite. When the CD4+ T-cell count falls below 100–200 cells per mm^3^, immunocompromised individuals typically experience a relapse of *Toxoplasma* infection [93]. New models have been created to study the recurrence of toxoplasmosis in mice, demonstrating the importance of IFN-γ in regulating the latent phase of the infection [94]. Recent research has clarified our understanding of how translational control contributes to the stress response and differentiation of the parasite. It is widely recognized that cellular stresses can activate the regulation of mRNA translation through the phosphorylation of the alpha subunit of eukaryotic translation initiation factor-2 (eIF2α). Once eIF2α is phosphorylated, it inhibits global translation initiation, allowing the selective translation of a specific group of mRNAs that code for proteins that mitigate the stress response [95]. 

## 4. Mechanisms of Persistence in *Trypanosoma* spp. 

*Trypanosoma* is a parasitic protozoan with a complex life cycle and can infect vertebrates worldwide. African trypanosomiasis, or sleeping sickness disease, is caused by trypanosomes of African species (*T. brucei rhodesiense* and *T. brucei gambiense*), which are transmitted by the tsetse fly and can infect other animal species [96,97]. The parasites enter first the lymphatic system and then the bloodstream and other body fluids where they can replicate extracellularly by binary fission. African trypanosomiasis causes a multitude of non-specific symptoms in the earlier period of infection and progressive confusion and other neurological problems when the parasites migrate into the central nervous system [98]. Over 55 million people are thought to be at various levels of risk of infection [96,99]. However, a consistent reduction in the rates of infection over the last two decades has now made it likely that the disease will be eliminated by 2030 [100]. On the other hand, Chagas disease, or American trypanosomiasis, is a multisystemic disease that can affect the cardiovascular, digestive, and central nervous systems [101]. This disease is caused by *T. cruzi*, a hemoflagellate parasite, transmitted by various hematophagous reduviid insects (kissing bugs), mostly in endemic areas [102,103,104]. The trypomastigotes invade the host cell where they differentiate into intracellular amastigotes. The amastigotes replicate and transform into trypomastigotes upon release into the bloodstream. The bloodstream trypomastigotes do not replicate extracellularly, in contrast to the African trypanosomes. *T. cruzi* is estimated to infect around 10 million people worldwide, primarily in Latin America. While the United States is not an endemic region, cases have been documented in southern states including Texas and Arizona [102,105].

Effective control programs for trypanosomiasis depend on a combination of vector control practices, timely patient diagnoses, and effective treatments, as there is no prophylactic vaccine available. Implementing these measures can significantly reduce the high burden of the disease [106]. The only approved drugs for treating American trypanosomiasis are benznidazole (BZN) and nifurtimox (NFX), which are nitroheterocyclic compounds. In the context of African trypanosomiasis, the primary treatment options consist of pentamidine for the initial stage and a combination therapy involving nifurtimox and eflornithine for the second stage of *T. brucei gambiense* infection. Conversely, the treatment regimen involves suramin for the first stage and Melarsoprol for the second stage in the case of the disease caused by *T. brucei rhodesiense* [107]. Although these drugs can clearly kill the parasite in humans, they should be used carefully during pregnancy and can cause adverse effects that result in treatment discontinuation in 15–20% of the patients [107,108]. 

The existence of persisters is a significant obstacle in eliminating *Trypanosoma* infections. For *T. cruzi*, these persisters are believed to originate from non-dividing amastigotes, which can form spontaneously both in vitro and in vivo due to stress-induced proliferation [109]. The persistence of these parasites is not attributed to drug resistance but rather to drug tolerance, as the new population of parasites that grows after drug removal does not exhibit a change in susceptibility to BZN in vitro, as compared to the original population [21]. Persistence was also observed as a result of DNA damage caused by gamma radiation or genotoxic agents, which was accompanied by an increase in transcription of TcRAD51, a protein responsible for the repairing of double-strand breaks (DSB) [110]

Sánchez-Valdéz and colleagues identified transiently dormant amastigotes as a key factor in drug treatment failure in patients with *T. cruzi* infection [21]. The study found that these dormant amastigotes exhibited resistance to trypanocidal compounds even after >30 days of drug exposure and could respond to cues that induced the conversion to trypomastigote forms within the host cells. This means that the dormant amastigotes have two potential outcomes: they may either re-initiate replication like conventional amastigotes or convert to trypomastigotes in the presence of other amastigotes that are undergoing this conversion process within the same host cells. Table 2 summarizes the transcriptome, metabolome, and translatome changes observed in *Trypanosoma cruzi* persisters.

Li et al. conducted a transcriptomic analysis to understand how *T. cruzi* persists inside the host cells [113]. They observed significant changes in the parasite’s mRNA load within the first four hours since the invasion of human fibroblasts. This indicated a dramatic environmental shift and the initiation of the amastigote differentiation program. The authors noted that extensive transcriptome remodeling was necessary to activate the amastigote differentiation program, leading to associated changes in morphology and functionality reflected in transcriptomic signatures. This included the downregulation of transcripts encoding major surface protein classes, flagellar assembly, and motility genes and the upregulation of amastigote-specific surface proteins, GPI-inositol deacylase, membrane-bound/secreted phospholipase A1, and surface-localized phosphatidylinositol-phospholipase C. The data also suggested signaling pathway retooling, with differentially expressed predicted protein kinases and phosphatases shortly after the trypomastigote entry into the mammalian host cells [116]. 

Despite *T. brucei* does not undergo differentiation into any intracellular stage within the vertebrate host, it was reported that this parasite is capable of colonizing the adipose tissue. The adipose tissue forms (ATFs) of the pathogen exhibit a slower rate of multiplication compared to their counterparts in the bloodstream, as well as aeduced protein synthesis. Intriguingly, this population of parasites can revert their proliferation profile in the blood and are also refractory to drug treatments (pentamidine and melarsoprol), which collectively contribute to the persistence of the disease they cause and treatment failure [115].

## 5. Mechanisms of Persistence in *Leishmania* spp.

*Leishmania* is a genus of parasitic digenetic protozoans transmitted by sandfly vectors and can cause a range of diseases known as leishmaniasis. The severity of these diseases varies from self-healing skin lesions to potentially fatal visceral leishmaniasis affecting internal organs [117,118]. Leishmaniasis is a neglected tropical disease that affects impoverished populations in more than 90 countries throughout Asia, Africa, the Middle East, and Central and South America. The incidence of cutaneous leishmaniasis (CL) is likely higher than reported, with estimates ranging from 700,000 to 1.2 million cases annually [119], mostly occurring in the Americas, the Mediterranean, the Middle East, and Central Asia [17]. Fewer cases of visceral leishmaniasis (VL) are reported annually, currently less than 100,000, with over 95% of them reported to the World Health Organization (WHO) from Brazil, China, Ethiopia, India, Kenya, Nepal, Somalia, and Sudan. The risk factors include poverty, migration, malnutrition, poor hygiene, and an immunocompromised health state [17,119]. 

The *Leishmania* parasite has over 20 characterized species and is transmitted by approximately 70 types of phlebotomine sandflies [120,121]. Chemotherapy is a key management strategy for leishmaniasis, with antimonials being the primary drugs used in several regions. However, these compounds have a narrow therapeutic window, and resistant *Leishmania* strains threaten their efficacy. Recently, our group uncovered that antimony drug resistance is established not only through the acquisition of genetic mutations, as shown in earlier studies, but through the reprogramming of translation that coordinates the lipidome and metabolome remodeling to counteract the drug [122,123,124]. Miltefosine, paromomycin, and amphotericin B are other compounds used to treat leishmaniasis, but they also have toxicity issues and face resistance challenges [125,126]. *Leishmania* is characterized by a life cycle that involves two distinct forms: intracellular amastigotes inside the mammalian host and motile promastigotes living extracellularly in the sand fly vector. During the infection of mammalian hosts, *Leishmania* transforms into intracellular amastigotes, which rely on phagocytes as the host cells [127]. After entering the mammalian host, the *Leishmania* promastigotes typically reside in macrophages, followed by transformation into amastigotes that replicate within modified phagolysosomes [128]. The amastigotes display a slower metabolism [129]. Metabolomic studies revealed that axenic amastigotes have a reduced level of multiple amino acids in comparison with promastigotes [130]. Two distinct populations of amastigotes were found using an infected mouse model: an actively dividing population and a non-dividing, dormant population [20]. The dormant population possessed certain characteristics such as reduced metabolism, altered host metabolism, and other changes at the molecular level contributing to its persistence during infections, as illustrated in Figure 3. 

To effectively infect its host, *Leishmania* parasites weaken the host’s antimicrobial responses, disrupt the normal vesicle trafficking, and manipulate the host’s immune and metabolic functions. This is accomplished by influencing various signaling pathways and transcription factors in the host cell at the molecular level [20]. Studies that profiled the mRNA levels in *L. donovani*-infected macrophages, showed a significant perturbation in the host gene expression programs associated with parasite persistence; specifically, the axenic amastigotes of *L. donovani* downregulated the expression of genes involved in apoptosis and NF-κB signaling while upregulating those encoding monocyte chemoattractants [131,132]. Furthermore, DNA-microarray-based studies revealed that infection with *L. donovani* promastigotes led to increased levels of transcripts related to cell migration and to the repression of genes encoding MHC class II molecules in human and mouse monocyte-derived macrophages [133,134]. A more recent RNA sequencing (RNA-seq) study of mouse peritoneal macrophages infected with *L. donovani* also showed a significant suppression of genes related to immune activation, signal transduction, phagosome, and endocytosis [132,134,135]. One way cells can adapt to changes in their environment is through selectively altering the translation efficiency of specific transcripts, a post-transcriptional mechanism that allows a rapid proteome remodeling without requiring the synthesis of new mRNA [136,137].

The *Leishmania* amastigotes have been found to demonstrate certain persister-like cellular states. A study showed that non-dividing intracellular amastigote populations could be clearly distinguished from their actively replicative counterparts [20]. According to a study comparing *L. (Viannia) braziliensis* promastigotes and amastigotes, the amastigotes demonstrated diminished levels of protein, RNA, mitochondrial kDNA, and ATP and no in vitro growth inside macrophages [130]. The ability of promastigotes and amastigotes to enter a quiescent state using stationary-phase organisms and drug pressure as models has been demonstrated recently [138]. The quiescent cells displayed a global reduction in both transcription and metabolism. A small subset of transcripts was upregulated and included amastins and GP63. GP63 is known to play a key role in the evasion and inactivation of the host’s innate immune system [139,140,141]. However, the molecular function of amastins and the role they play in quiescence remain poorly understood.

Recent research using transcriptome, ribosome, and polysome profiling showed that the parasite also selectively modifies the host cell’s proteome by regulating the translation efficiency of specific subsets of mRNAs [133]. It also represses the host translation through mTOR cleavage by GP63 [140]. Moreover, during *L. donovani* infection, the translation of mRNAs that encode MHC class I components is suppressed, potentially leading to dysregulated antigen presentation [142], thereby sabotaging the host immune response. On the other hand, during initial infection, the translation of mRNAs related to chromatin remodeling, such as histones and DNA/histone-modifying enzymes, is increased, suggesting a mechanism by which the parasite directs epigenetic changes that inhibit the host cell’s innate immune response [143]. Likewise, a proteomic study found that several histones and chromatin-remodeling proteins were induced in macrophages during *L. donovani* infection, which correlated with increased transcriptional activity [144]. Currently, it remains unknown if persisters use the same mechanisms to escape the host immune response as non-dormant cells. 

As *Leishmania* parasites lack transcriptional regulation for gene expression, translational control becomes crucial for adaptive responses during their transformation from promastigote to amastigote forms. This process is accompanied by a reduction in mRNA translation, which is associated with eIF2α alpha-subunit phosphorylation. Eukaryotes typically undergo global translation reduction in response to stress, and phosphorylation of eIF2α is a major pathway involved in this stress response, particularly during *Leishmania* persistence in the vertebrate host [145]. Currently, it is poorly understood how translational control supports the persistence mechanisms during parasite transformation from a proliferating to a quiescent state. This is one of the most critical questions that needs to be addressed in the future investigations.

## 6. Concluding Remarks

Persistent cells of protozoan parasites have certain common characteristics that facilitate their survival and persistence within their hosts. The observed characteristics encompass the capacity to adapt to dynamic host surroundings, modifying their metabolic pathways to economize resources and circumvent the host’s immune system. Protozoan parasites have developed diverse molecular mechanisms to accomplish the adaptations they undertake. The transformation to the state of persistence includes a global and coordinated reduction in DNA replication, transcription, translation, and metabolism as well as the manipulation of host signaling pathways. It is quite likely that additional investigation into the molecular mechanisms that govern persistence will generate novel perspectives in the biology of these pathogens and, consequently, streamline the development of groundbreaking therapeutic strategies. The persistence of protozoan parasites within their hosts is a complex process that involves a variety of molecular mechanisms. These mechanisms enable the parasites to evade the host immune response, establish long-term infections, and survive in hostile environments. Persisters are known to exist for other protozoa species such as *Balamuthia mandrillaris* and *Acanthamoeba* spp., which may form dormant cysts under harsh conditions [146,147]. *Acanthamoeba* is a free-living protozoan that can cause acanthamoeba keratitis. Its double-walled dormant cysts can survive antibiotics, low temperatures, high-dose UV and γ-radiation. The capacity of protozoan parasites to persist within their hosts is a significant aspect that contributes to their pathogenicity and their potential to instigate persistent infections. Understanding the molecular mechanisms underlying persistence is of utmost importance for the development of effective therapies and vaccines to counteract such infections. 

Protozoan persistence has been the subject of intense research in recent times. The advent of advanced techniques in molecular biology, transcriptomics, and proteomics has enabled researchers to gain valuable insights into the underlying mechanisms that facilitate this phenomenon. The scientific community is currently engaged in a concerted effort to advance our knowledge of the molecular mechanisms underlying the persistence of protozoan parasite cells. Despite significant strides in this field, there remain several critical research gaps that require further investigation. Several domains necessitate further exploration. For example, (I) the identification of novel genes and proteins involved in persistence, as there are probably numerous genes and proteins involved in persistence that are yet unknown, despite developments in genome sequencing and proteomics. For a thorough knowledge of the molecular mechanisms driving persistence, identifying these molecules and their roles is crucial; (II) understanding the interplay between host and parasite during persistence; the host immune system’s influence on the molecular mechanisms of persistence is crucial. To better understand how protozoan parasites interact with the host cells and control the host immune responses to promote their own survival, further investigations are needed; (III) studying the effects of environmental factors on persistence; the microenvironment within the host can vary widely, and different environments may exert varying effects on the molecular mechanisms of persistence. Further studies are needed to understand how environmental factors, such as nutrient availability and oxygen levels, affect the persistence of protozoan parasites.

By addressing these research gaps, we could better grasp the biology of persistent protozoan diseases, thus contributing to developing novel therapeutics and improving our overall understanding of the biology of the protozoan persisters. As our understanding of these mechanisms deepens, we might anticipate developing new strategies for intervening in the processes involved in parasite persistence, which will ultimately lead to better treatments and outcomes for those affected by these parasitic infections. 

## Figures and Tables

**Figure 1 microorganisms-11-02248-f001:**
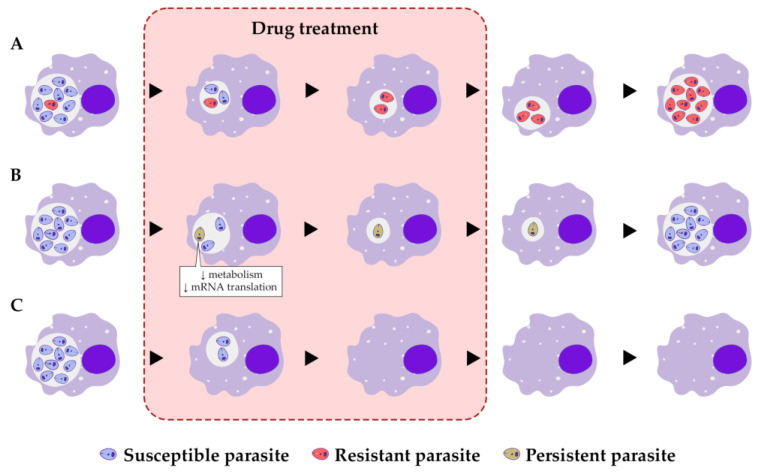
Types of response of protozoan parasites to a drug treatment. The responses of the parasites are shown in different periods of time during the drug treatment. (**A**) Response of resistant parasites. Susceptible parasites (purple) perish because of the drug treatment; in contrast, due to inherited mutations or molecular mechanisms such as translational reprogramming, resistant parasites (red) can survive and proliferate. (**B**) Response of persister-like parasites. A persister-like parasite (yellow) enters a dormancy state in the presence of the drug or other stresses and resumes proliferation only when the treatment is finished or the stress is removed. (**C**) Response of susceptible parasites. Susceptible parasites (purple) lack mechanisms to avoid the effects of the drug and die during the treatment.

**Figure 2 microorganisms-11-02248-f002:**
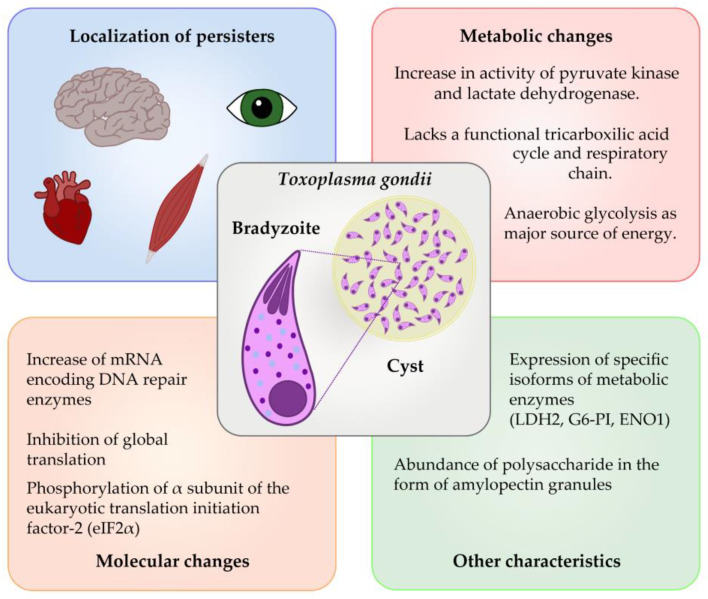
Characteristics of persister-like cells of *Toxoplasma gondii.* The bradyzoites are the persister-like form of *T. gondii*, which are encysted in different tissues, like the brain, the eye, the skeletal muscle, and the cardiac muscle. As with other protozoa parasites, molecular and metabolic changes are involved in the dormancy state of *T. gondii*.

**Figure 3 microorganisms-11-02248-f003:**
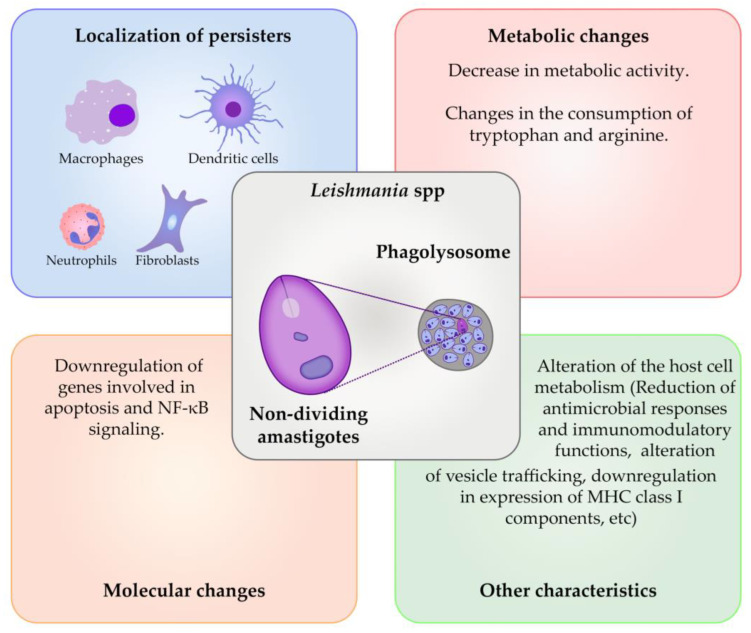
Characteristics of persister-like cells of *Leishmania* spp. The amastigote could be present in the non-dividing persister form of *Leishmania* spp., which could be found in the phagolysosomes of phagocytic cells, such as macrophages, dendritic cells, neutrophils, and fibroblasts. The reduction of the metabolic activity is one of the main characteristics of *Leishmania* spp., as well as the alteration of the host cell’s metabolism.

**Table 1 microorganisms-11-02248-t001:** Features of *Plasmodium* persisters.

Traits	Description	References
Location of the dormant cells	Host’s red blood cells and hepatocytes (*P. vivax*, *P. ovale*)	[48,66,67,68]
Transcriptome changes	Stage-specific gene expression, variant surface antigens (VSAs) expression (including the *var* gene family, which encodes the erythrocyte membrane protein 1 PfEMP1), utilizing non-coding RNAs, inducing stress-responsive pathways. Overexpression of genes encoding specific cellular RBPs and proteases like Vivapains	[33,34,49]
Translatome changes	DOZI binding to the ribonucleoprotein complex, with translation repression. Phosphorylation of eIF2α and formation of stress granules. Translational repression	[41,58,59,63,64,65]
Metabolomic changes	Metabolic activity decrease, nutrient uptake decrease, and active apicoplasts and mitochondria. Restructuring of mitochondria–nucleus interaction.	[43,46,57]

**Table 2 microorganisms-11-02248-t002:** Features of *Trypanosoma cruzi* persisters.

Traits	Description	References
Location of the dormant cells	Blood, lymph, and subcutaneous tissues (particularly the cardiac muscle) and CNS	[111,112]
Transcriptomic changes	Downregulation of transcripts of major polymorphic surface proteins, reduced expression of genes coding proteins involved in flagellar assembly and motility, shortening of the single T. cruzi flagellum; increased abundance of δ-amastin; increased transcript levels of GPI-inositol deacylase, membrane-bound/secreted phospholipase A1, and surface-localized phosphatidylinositol-phospholipase C (PI-PLC). Upregulation of proteins in charge of DSB repairing	[110,113,114]
Proteome changes	Differential expression of plasma membrane proteins, protein kinases, and phosphatases. Reduced protein synthesis in adipose tissue (*T. brucei*)	[113,115]
Metabolome changes	Signaling pathway retooling with differentially expressed protein kinases and phosphatases.	[116]

## Data Availability

Not applicable.

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
