# Peer review of "Molecular Mechanisms of Persistence in Protozoan Parasites"

_microorganisms, 2023, doi:10.3390/microorganisms11092248_

Round 1

Reviewer 1 Report

The review manuscript by Tarannum et al. entitled “Molecular mechanisms of persistence in protozoan parasites” focusses on the protozoan parasites Plasmodium, Toxoplasma, Trypanosoma and Leishmania. After a well written introduction on persister-like cells and its impact on treatment of pathogens, the authors discuss the current knowledge on persistence for those 4 protozoan parasites. Although the novelty of the manuscript is limited, as no new insights on the molecular mechanisms is provided, the review does provide a comprehensive overview of the current knowledge on persistence in protozoan parasites. The following comments should be addressed.

1)      In chapter 4 both African trypanosomiasis and American trypanosomiasis (Chagas disease) are introduced (lines 338-357). From line 358 onwards the authors use the term “trypanosomiasis” without specifying whether they refer to T. brucei or T. cruzi. However, most statements apply to T. cruzi only (e.g. drugs lines 360-364; experimental persister cell data 36-370 and dormancy 371-379). As no information is provided on persister cells for T. brucei, the authors should rewrite this entire section and substantially shorten the introduction about T. brucei to ensure that it is clear that this chapter is about T. cruzi and not about T. brucei spp.

2)      In chapter 2 the authors discuss persistence in Plasmodium and the authors correctly focus on hypnozoite stages in the liver and on the slow developing asexual stages in the erythrocytes. The authors do not discuss the sexual stages of the parasite in erythrocytes despite the fact that these stage are much longer persisting in the bloodstream of their host (weeks!) than the asexual stages (days). The authors are requested to include the gametocyte stage in this chapter as well.

3) Next the four discussed parasitic protozoa, the free-living amoeba Acanthamoeba and Balamuthia mandrillaris also have dormant cyst stages that occur within the host. For amoebic keratitis this is clinical relevant and this type of dormant stages could be discussed by the authors as well.

Minor points

1)      Line 221. Change establish infection into induce disease (or symptoms). Patients are already infected by the Plasmodium parasites once the sporozoites are introduced by the bite the mosquito vector.

2)      Line 258. Not only HIV patients are susceptible for reactivation of latent toxoplasmosis, so are allogenic human stem cell transplant patients.

Reviewer 2 Report

Parasitic protozoa, such as Plasmodium species that cause malaria, Toxoplasma gondii and Kinetoplastid protozoa, including Trypanosoma cruzi and Leishmania spp., cause millions of deaths globally. These organisms can evolve drug resistance and they also exhibit phenotypic diversity, including the formation of quiescent or dormant forms that frequently results in prolonged infections and the recurrence of diseases. In this manuscript, the authors provide a comprehensive overview of the latest developments investigating protozoan persistence in vertebrate hosts. The focus is primarily on the function of persisters, their formation within the host, and the specific molecular interactions between host and parasite while they persist. The work has a significant contribution to the field. In most cases, the work is well organized and comprehensively described. However, there are several comments and suggestions as follow.

1. Some mechanisms of persistence may be shared by different protozoa parasites. It should be presented as a whole picture or table. While other mechanisms may be unique to one or two parasites, it should be illustrated separately. The relationship between different mechanisms and life cycles of different parasites should be discussed in more detail.

2. Both figures and tables were used to illustrate the mechanisms of different protozoa parasites. But they are not completely cover the relative content in the manuscript. In table 1 and table 2, the cited references are not enough and more references should be added.

3. The background of four protozoa parasites should be more concise. The specific molecular interactions between host and parasite while they persist need for further enrichment and organized.

Round 2

Reviewer 1 Report

Comments have been addressed properly.